# SCAPE: Learning Stiffness Control from Augmented Position Control Experiences

**Mincheol Kim**[1], **Scott Niekum**[2], **Ashish D. Deshpande**[1]
[1]Department of Mechanical Engineering
[2]Department of Computer Science
The University of Texas at Austin, USA
`mincheol@utexas.edu, sniekum@cs.utexas.edu, ashish@austin.utexas.edu`

**Abstract:** We introduce a sample-efficient method for learning state-dependent stiffness control policies for dexterous manipulation. The ability to control stiffness facilitates safe and reliable manipulation by providing compliance and robustness to uncertainties. Most current reinforcement learning approaches to achieve robotic manipulation have exclusively focused on position control, often due to the difficulty of learning high-dimensional stiffness control policies. This difficulty can be partially mitigated via policy guidance such as imitation learning. However, expert stiffness control demonstrations are often expensive or infeasible to record. Therefore, we present an approach to learn Stiffness Control from Augmented Position control Experiences (SCAPE) that bypasses this difficulty by transforming position control demonstrations into approximate, suboptimal stiffness control demonstrations. Then, the suboptimality of the augmented demonstrations is addressed by using complementary techniques that help the agent safely learn from both the demonstrations and reinforcement learning. By using simulation tools and experiments on a robotic testbed, we show that the proposed approach efficiently learns safe manipulation policies and outperforms learned position control policies and several other baseline learning algorithms.

**Keywords:** Manipulation, Stiffness control, Reinforcement learning

## 1  Introduction

In recent years, deep reinforcement learning has been successfully used to improve object manipulation with robotic hands [1, 2]. However, one of the primary limitations of these works is that in most robotic hands, the robot joint poses are explicitly controlled through position control and forces are implicitly decided. Lack of explicit control over the forces leads to limited safety and inability to handle uncertainties [3]. These might be critical factors when a robot is operating in unstructured environments and during the exploratory phase of the learning process [4, 5].

Modulation of stiffness in concert with position control has been shown to address robustness, safety, and performance under uncertainties [6, 7, 8], and has gained much attention in the learning community as well [9]. However, stiffness control imposes additional action dimensions, which affects sample-efficiency of policy learning. This hindrance can be partially mitigated through guidance via imitation learning [10]. Expert demonstrations have been successfully collected and used in policy learning for position control-based robotic hands [1, 11, 12]. However, in the case of stiffness control, such expert demonstrations are expensive and difficult to acquire. Typical demonstrations can directly be recorded via various sensors, but stiffness is not a measurable quantity, but rather a relationship.

In prior literature, admittance control has been used to capture equilibrium position trajectories [13, 14]. Subtle and quick impact perturbations are used to measure the compensatory forces and torques employed by the demonstrator while performing the trajectory, which are then used to implicitly calculate the stiffness at certain positions. These stiffness estimates are then used to further estimate and model the stiffness profiles, thereby compounding potential errors. This estimation process is more ambiguous for object manipulation due to the required precision and accuracy, and therefore poses a major challenge to learning stiffness control from demonstrations.

5th Conference on Robot Learning (CoRL 2021), London, UK.

In this paper, we present a novel learning strategy—Stiffness Control from Augmented Position control Experiences (SCAPE)—for learning state-dependent stiffness control policies in high-dimensional problems such as dexterous manipulation. Imitation learning is used in conjunction with reinforcement learning to provide policy guidance, and we propose a way to bypass the need for stiffness demonstrations through an augmentation process. This process leverages the knowledge of the robot model such that we do not require expert stiffness control demonstrations. Instead, position control demonstrations are augmented to infer approximate, suboptimal stiffness control demonstrations. To address this suboptimality, we use a Q-filter [11] to prevent the agent from mimicking dangerous choices that may appear in the inferred stiffness demonstrations. We also introduce the concept of an imitation regulator that controls the mode of imitation depending on the assessment of the current policy. Ablation studies show that these techniques play meaningful roles in both safety and stability of learning. Through simulation and experiments, we show that SCAPE produces a successful policy that is robust to different types of realistic uncertainties, and safe in terms of force interaction.

## 2 Background and Related Works

Some of the notable past works in learning stiffness control rely on a reference trajectory, which we refer to as trajectory-dependent approaches. However, the lack of robustness to uncertainties and variability in the task dynamics renders these approaches inapplicable to dexterous manipulation. Other recent research aims to learn a stiffness controller that dynamically adapts to the environment, which we refer to as state-dependent approaches. In this section, we briefly explain some of the notable attempts to learn stiffness control, and discuss possible shortcomings.

### 2.1 Trajectory-dependent Stiffness Controllers

One possible approach is to learn time-indexed gain scheduling through $PI^2$ [15], which is a stochastic optimization method that results in a time-indexed reference trajectory that can be tracked by the robot without modeling the inverse kinematics or dynamics. This approach adds additional parameters to control compliance so that the resulting controller takes environment dynamics into account and modulates the gains accordingly [16, 17, 18]. However, a solution from $PI^2$ can only optimize about a pre-defined cost function and cannot be used for object-centered manipulation, which requires highly divergent position and stiffness trajectories depending on the goal and observations.

Another approach uses an Incremental Gaussian Mixture Model (IGMM) and Gaussian Mixture Regression (GMR) to predict the interaction force for the next time-step, and feed-forward appropriate control effort [19, 20]. The goal of this approach is to learn a feed-forward model such that the feedback stiffness control effort can be minimized. However, this approach makes a critical assumption that expert demonstrations with force trajectories are available, which renders it inapplicable without such demonstrations.

Trajectory planners combined with reinforcement learning can also be used. Once the trajectory is defined, a residual control policy can be learned to adjust the gains according to the current observation. This method is mostly used in simple tasks where a trajectory planner is readily available, such as in peg-in-hole assembly tasks [21, 22, 23]. While this is a suitable approach for closed environments, it is less effective for dexterous manipulation where desired trajectories can change based on dynamic observations such as dropping the object or unexpected interaction forces. In addition, due to the lack of policy guidance, the learning process requires a complex reward function as well as an extensive amount of training time even with a trajectory planner aiding the policy search.

### 2.2 State-dependent Stiffness Controllers

Due to the specificity of the solution, relying on a fixed reference trajectory or scheduled gain is bound to result in catastrophic failure in dynamically changing environments. To account for a large degree of variability in the environment, state-dependent stiffness control policies have recently been proposed. In this paper, we compare our work with these state-dependent methods, as the existing trajectory-dependent approaches are inapplicable in dynamic settings.

In one related approach, during hopping and wiping motions [24], the robots successfully produce stiffness control policies that outperform direct torque control and position control policies. How-

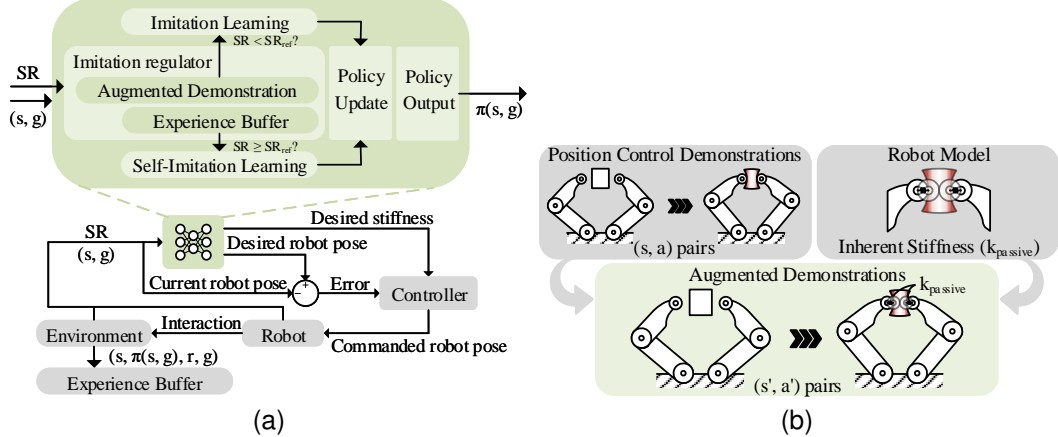

Figure 1: (a) Proposed learning scheme for SCAPE. (b) Proposed augmentation process of position control demonstrations.

ever, this approach is strictly limited to simple and repetitive low-dimensional movements. For instance, the robot is allowed to move only in one direction in the hopping task, or along a pre-defined circular path in the wiping task.

A similar work demonstrates the performance difference based on the form of the control policy [25], where variable impedance control outperforms all other controller types in simple tasks. Despite the effort to simplify the problem into a lower-dimensional manifold by keeping the gripper closed in the door opening task, the variable impedance control (VIC) fails to outperform the fixed impedance control. A possible culprit can be the high dimensionality induced by VIC. A similar work [26] observes stability-guaranteed learning for VIC during peg-in-hole tasks.

Other work depicts a task-space impedance controller's performance in quadruped locomotion [27]. Variable impedance control policies outperform the direct torque control policy in terms of cumulative rewards and robustness to disturbances. Interestingly, impedance controllers resulted in more energy-efficient policies although the reward function did not consider energy. It is worth noting that an extensively tuned gait planner was provided to the agent to learn a successful policy.

While these state-dependent approaches show promising results in terms of robustness to uncertainties, these existing approaches focus on simple single-stage, repetitive tasks and require extensive reward shaping due to the lack of policy guidance. In the next section, we explain how SCAPE addresses this issue and produces successful state-dependent stiffness control policies, which can be used in multi-stage tasks such as grasping and manipulation.

## 3   Methods

In this section, we present our approach, SCAPE, to learn state-dependent stiffness control without requiring stiffness control demonstrations. Use of imitation learning within reinforcement learning improves policy guidance, thereby enabling high-dimensional dexterous manipulation. For SCAPE and all the baselines, we employ Deep Deterministic Policy Gradient with Hindsight Experience Replay (DDPG + HER, [11]). The overall learning scheme is depicted in Fig. 1a, where the stiffness control policy takes in observed states $s$ and the goal $g$, and produces an action $\pi(s, g)$ which contains the desired stiffness in addition to the desired pose. The controller block is the high-level controller, which employs task-space stiffness control. SR refers to the overall success rate of the current policy (i.e., how often does the object reach the goal states while staying intact?).

A commonly used action representation in dexterous manipulation describes only the desired position of the actuator in the inner control loop, hence the name position control. On the other hand, a stiffness control policy outputs actions that describe the desired position of the actuators as well as the desired stiffness in the corresponding joint, as shown in Fig. 1a. However, as explained earlier, it is difficult to obtain expert stiffness control demonstrations for imitation learning. Therefore, we use augmented position control demonstrations as shown in Fig. 1b.

These demonstrations are typically generated from teleoperation or kinesthetic teaching, and contain desired position trajectories. In our study, we use 25 demonstrations that accomplish task-related kinematic goals without consideration for the object fragility (e.g., commanded to fully close the grippers). We leverage the fact that the stiffness of the robot is known either from the simulation model or the hardware specifications, and that position control works by moving to the desired position with the inherent stiffness or position gain of the actuator. In this paper, we refer to this inherent stiffness as $\mathbf{k}_{passive}$. Therefore, state-action pairs, $(s, a)$, in common position control demonstrations can be augmented to that of stiffness control demonstrations, $(s', a')$, where the desired stiffness is $\mathbf{k}_{passive}$. Consequently, these augmented demonstrations become suboptimal stiffness control demonstrations since the commanded stiffness of the robot is fixed to $\mathbf{k}_{passive}$. We can then infer the reward function of the task from the augmented demonstrations without manual reward shaping, and at the same time use it to learn improved stiffness control. Note that $\mathbf{k}_{passive}$ is n-dimensional stiffness, from which we can choose any dimension of interest and modulate the stiffness. In this paper, we modulate the stiffness in the grasping dimension (i.e., $\mathbf{k}_{passive} \in \mathbb{R}$).

## 3.1 Outperforming the Demonstration

Simple imitation learning in the form of behavioral cloning does not allow the agent to improve beyond the performance of the demonstrations due to the cloning loss [11]. The weight of the cloning loss can be decreased iteratively, assuming that the agent is able to learn a policy equivalent to the demonstrator early in the iteration [1]. But it is unclear how to determine the amount of dependency on the demonstrations with respect to the iteration. Also, simply reducing the dependency does not prevent the agent from cloning the undesirable behaviors seen in the suboptimal demonstrations we use. Therefore, we adopt additional techniques to encourage the policy to outperform the augmented demonstrations and address its suboptimality.

### 3.1.1 Q-Filter

We use a Q-filter [11] to choose which replay transitions to clone from. The fundamental motivation behind learning from demonstrations is the assumption that the demonstrations provide a near-optimal action. However, this is not true for the augmented demonstrations. A Q-filter allows the agent to compare the Q values produced by the transitions from demonstrations, $(s_i, a_i, g)$, and the current policy, $(s_i, \pi_\theta(s_i, g), g)$. By comparing their values, the agent does not clone the behavior if its current policy provides a better action for the given demonstration state. More formally, the cloning loss $L_{bc}$ can be defined as:

$$L_{bc} = ||a_i - \pi_\theta(s_i, g_i)|| \mathbb{1}_{Q(s_i, a_i, g_i) > Q(s_i, \pi_\theta(s_i, g_i), g_i)} \tag{1}$$

However, it is often difficult to infer the subtle difference in the qualities of the policies solely from examining the resulting Q estimates due to the overestimation issue of Q values [28]. Although the usage of the Q-filter improves the safety of learning, it tends to produce oscillatory gradients that prevent convergence of the policy due to its Boolean property [11].

### 3.1.2 Imitation Regulator

To improve convergence of our method, we introduce an imitation regulator that observes the overall success rates of the current policy and determines the appropriate source of imitation from: 1) the augmented demonstrations and 2) the agent's own past experience. The latter is sometimes referred to as self-imitation learning [29]. The regulator controls the replay buffer $\mathcal{D}_{IR}$ used for sampling transition batches to imitate as follows:

$$\mathcal{D}_{IR} = \begin{cases} \mathcal{D}_{demo}, & \text{if } SR < SR_{ref} \\ \mathcal{D}_{SIL}, & \text{otherwise} \end{cases} \tag{2}$$

where $\mathcal{D}_{SIL}$ and $D_{demo}$ refer to the buffers that store the actual experience replay and the augmented demonstrations, respectively. $SR \in [0, 1]$ is the overall success rate of the current policy. It is considered an overall success if the object reaches the goal states and also stays intact throughout the episode. $SR_{ref}$ is the reference success rate that is empirically found. Put simply, the regulator actively switches the source of demonstrations from $\mathcal{D}_{demo}$ to $\mathcal{D}_{SIL}$ once the success rate reaches $SR_{ref}$. This brings three primary benefits. First, the agent no longer references the suboptimal demonstrations which contain undesirable behaviors. Second, the policy converges faster from the minimized oscillation of gradients. Third, by cloning the previously generated actions, the agent leverages exploration, thereby improving upon its current policy.

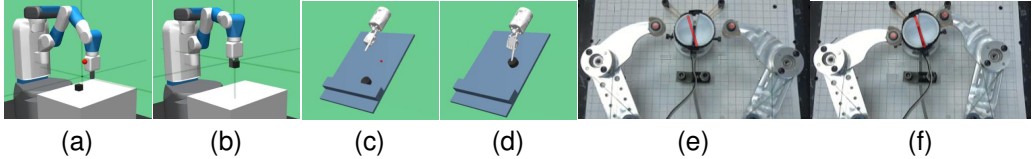

Figure 2: Initial and final scenes of the (a, b) Block, (c, d) Chip, and (e, f) NuFingers environments.

## 4 Environments

We use three robotic manipulation environments shown in Fig. 2 for simulation and experiments. Details for each environment can be found in Appendix A. For simulation, we use robots provided by OpenAI Gym [30]. In all environments, we do not use the ground-truth force measurements during training. Instead, we use quasi-static force measurements from the series elasticity of the robot, which do not require force sensors (i.e., $\mathbf{F} \approx \mathbf{k}_{passive}(\mathbf{x}_{desired} - \mathbf{x}_{current})$). Note that $\mathbf{x}$ is defined in the same coordinate frame as $\mathbf{k}_{passive}$. For safety evaluation however, unless stated otherwise, we use ground-truth force measurements. A successful policy must meet both task-related (e.g., did the object reach the goal states?) and safety-related (e.g., is the object intact?) goals. For instance, even if the object reaches the goal states, the task is considered a failure if the applied force exceeds the breaking force. The task-related kinematic reward is $r_{task}(s) : s \rightarrow \{-1, 0\}$, and the safety-related reward is $r_{safety}(s) : s \rightarrow (-\infty, 0]$. Combining these reward functions naturally leads the agent to accomplish the kinematic goal while minimizing the estimated interaction force. For all environments, the immediate reward $R$ for the observation $s$ is defined as:

$$R(s) = r_{task}(s) + r_{safety}(s)$$
$$r_{task}(s) = \begin{cases} 0, & \text{if kinematic goal is met} \\ -1, & \text{otherwise} \end{cases} \tag{3}$$
$$r_{safety}(s) = -\alpha ||\mathbf{F}|| - \beta ||\dot{\mathbf{q}}||$$

where $\mathbf{F}$ is the estimated force, $\dot{\mathbf{q}}$ is the joint velocity, and $\alpha$ and $\beta$ are normalization coefficients that depend on the environment.

Furthermore, we implement a low-pass filter with a time constant of $0.05s$ for all force measurements in the simulation. To validate the robustness under realistic conditions, we include three types of uncertainties during training: 1) random perturbation to the object within grasp, 2) measurement noise in the object's position, and 3) random control failure, that repeats the previous action.

### 4.1 Quasi-static Pick-and-place Environment (Block)

The Block environment entails a pick-and-place task and is used to verify whether SCAPE is capable of learning a safe and robust manipulation policy under quasi-static assumptions. In this environment we assume that the ground-truth force matches the estimated force, which is reasonable to assume when the object is in grasp and the involved masses are small enough. The observation includes relative positions between the object, gripper, and the goal, as well as the gripper configurations, estimated force, and stiffness. The action includes Cartesian movement of the gripper, change in the gripper configurations, the changes in stiffness and its limit.

### 4.2 Dynamic Pick-and-place Environment (Chip)

Chip environment is a dynamic version of the pick-and-place environment, designed to demonstrate that SCAPE produces a successful manipulation policy even in dynamic situations where the agent does not have access to ground-truth force measurements. The robot is required to slide the object up the wall using friction. Thus, the ground-truth force comes from not only the finger, but also from friction, which depends on velocity and normal force. The observation and action spaces are similar to the Block environment, but the observation also includes the fingertip velocity in Cartesian space, which is part of the kinematic goal in this case. If only position is considered for the kinematic goal, we have found that the agent constantly moves the object around the goal location. We postulate that this phenomenon arises from kinetic friction being smaller than static friction.

### 4.3 In-hand Manipulation Environment (NuFingers)

To demonstrate the applicability for in-hand manipulation, we use the NuFingers testbed [7]. We first train the agent with domain randomization [2] in a representative environment on Gym, and directly transfer the policy to the robot without any fine-tuning to validate its transferability and robustness under uncertainties. The ground-truth force measurements are only used for validation. The observation includes polar coordinates and relative orientation from the object of each finger, forces, joint velocities, and stiffness. The action includes define radial and tangential movements of the fingers. Stiffness modulation is applied in the radial direction, which dominates the grasping force. The task-related kinematic goal is to rotate the object to the desired orientation.

## 5 Results

In this section, we demonstrate the performance of SCAPE in various environments. Without the proposed augmentation process, baseline algorithms must learn a stiffness control policy from scratch [24, 25, 27], since expert demonstrations are not available. We compare SCAPE with these approaches for the ablation study in Sec. 5.2, as learning from scratch fails catastrophically. For the main experiments in Sec. 5.1, we compare our results with position control (existing approach), so that the difference only lies in the policy parametrization. This comparison demonstrates the importance of using state-dependent stiffness controllers when force-sensitive tasks are involved, as opposed to existing position counterparts that are widely used in dexterous manipulation [1, 2].

### 5.1 Experimental Results

Figure 3 depicts the success rates over epochs for the Block, Chip, and NuFingers environments. For in-depth assessment, we break down the plots to also show success rates for the safety and kinematic goals. Safety-related success rates refer to the fraction of the time when the experienced force is smaller than the breaking threshold. Note that exceeding it at any time results in an overall failure in the corresponding episode.

Kinematic pick-and-place tasks without uncertainties have been easily solved by the position control policies in previous works [31]. However, Fig. 3a shows that for position control policies, the heavy force penalty from large forces discourages exploration and prevents the agent from learning to even grasp the object. SCAPE on the other hand, uses stiffness control policies to successfully minimize the interaction force and reaches a 100% overall success rate.

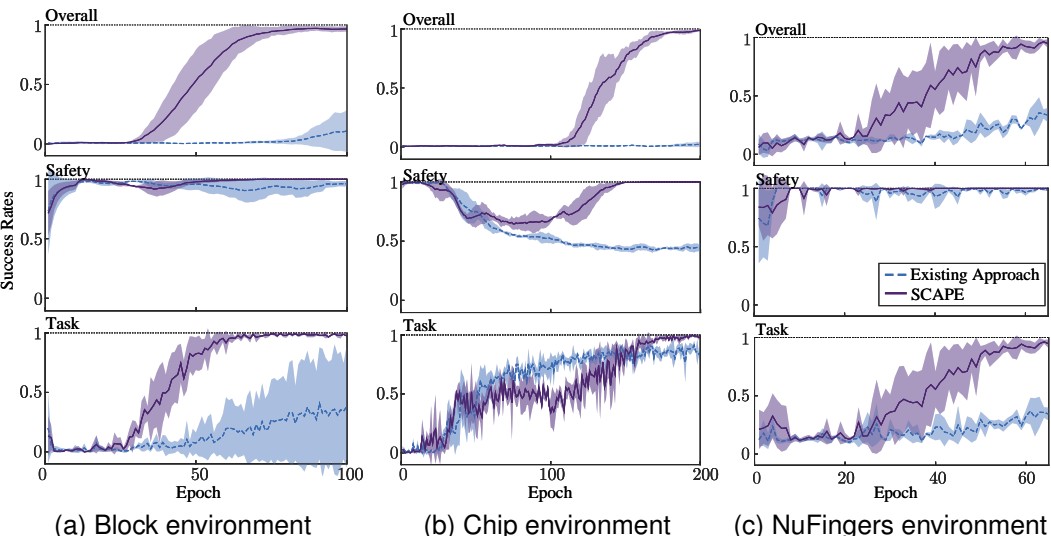

Figure 3: Resulting success rates of SCAPE (solid) compared to position control (dashed). Success rates for task-related goals (e.g., did the object reach the target states?) and safety-related goals (e.g., how often did the object stay intact?) are separately plotted. Overall goals entail both goals.

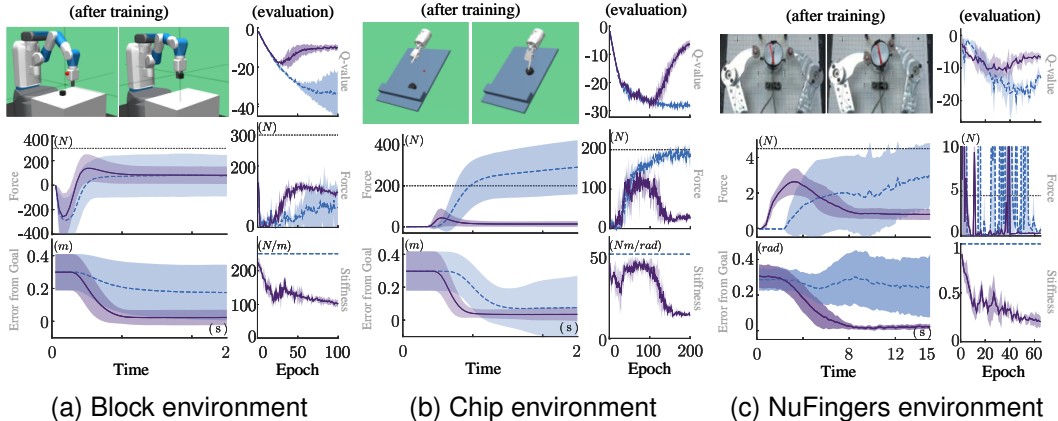

Figure 4: For each of the environments, force experienced by the objects and kinematic error of the resulting policies are shown on the left. The mean Q values, forces, and stiffnesses evaluated after each epoch are shown on the right.

For the Chip environment where the action space is smaller, the position-controlled agent learns to complete the kinematic task entailed in the demonstrations to some degree as shown in Fig. 3b. However, it fails to address the safety issues, thereby breaking the object in almost all of the evaluation episodes. Note that unlike in the Block environment, the ground-truth force measurements are used for evaluation. It is notable that SCAPE still produced a successful policy by referencing only the quasi-static force measurements from the series elasticity, which do not include friction.

For the NuFingers environment in Fig. 3c, we find a similar trend as in the Block environment. The position control approach fails to learn a policy that completes the kinematic task. SCAPE however, reaps the benefits of stiffness control and finds a successful policy. Most importantly, in spite of the model discrepancies between the simulation and the actual robot, the resulting policy proves to be successful after the sim-to-real transfer without additional training. To summarize, it is evident that a state-dependent stiffness control policy outperforms the position control policy in terms of safety and robustness under uncertainties and that the SCAPE is capable of producing successful policies.

Figure 4 depicts various data during and after training. It is evident that the proposed approach quickly learns the necessary stiffness for the completion of the kinematic task. The stiffness and interaction force of the system are strongly related to one another as can be seen from the similar trends of the two curves. The existing approach which uses position control on the other hand, does not adjust the stiffness and therefore fails to reduce the interaction force.

Also, notice that the objects in the Chip and NuFingers environments experience much more force than what the robot can exert. This is because for evaluation, we use the ground-truth forces which come from various sources, such as the friction that depends on the normal force and the velocity of the object. The proposed approach successfully minimizes the ground-truth force without the actual measurement. While actual measurements will improve the stability, we expect the improvement to be marginal in the tested environments since large masses or explosive movements are not involved.

## 5.2 Ablation Study

To examine the effects of each technique used in this paper, we examine the Block environment under five different conditions:

- Condition 1: Reinforcement learning from scratch [24, 25, 27], without a Q-filter, without an imitation regulator
- Condition 2: Reinforcement learning + imitation learning from augmented demonstrations, without a Q-filter, without an imitation regulator
- Condition 3: Reinforcement learning + imitation learning from augmented demonstrations, without a Q-filter, with an imitation regulator

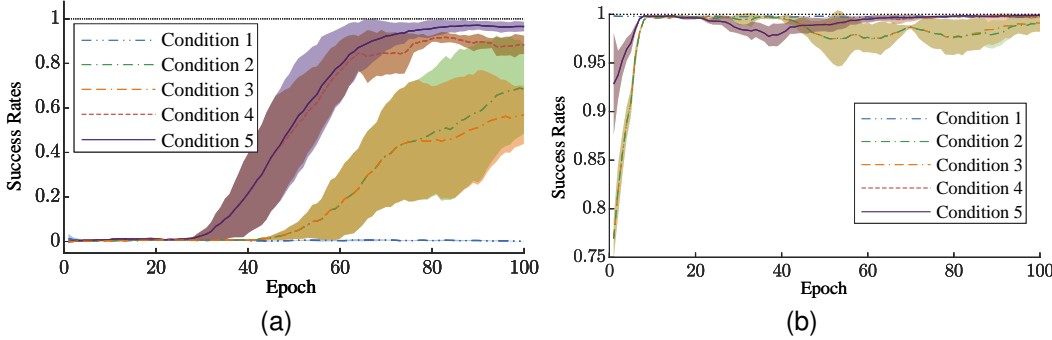

Figure 5: (a) Overall success rates during evaluation and (b) Safety-related success rates during exploration for different conditions in the Block environment.

- Condition 4: Reinforcement learning + imitation learning from augmented demonstrations, with a Q-filter, without an imitation regulator
- Condition 5: Reinforcement learning + imitation learning from augmented demonstrations, with a Q-filter, with an imitation regulator (SCAPE)

Condition 1 represents the approach taken by the researchers in the past [24, 25, 27], where imitation learning was not used due to the absence of demonstration data. Conditions 2-5 use the augmented demonstrations introduced in this paper, with different combinations of complementary techniques. Note that condition 5 is used to produce the results in Fig. 3.

Overall success rates generated under the different conditions are shown in Fig. 5a. From these results, we confirm that existing approaches [24, 25, 27] do not produce any meaningful results for multi-stage tasks (e.g., approaching an object, grabbing the object, and relocating the object). For such tasks, the augmented demonstrations play a crucial role in providing guidance to the policy through imitation learning.

Moreover, in conditions 2 and 3, the agent is unable to filter out undesirable behaviors, thereby consistently breaking the object during exploration as shown in Fig. 5b. Therefore, we confirm that the Q-filter allows the agent to make safe decisions. Also, applying the imitation regulator without the Q-filter fails from satisfying safety goals. Lastly, without the imitation regulator, the agent keeps referencing the suboptimal demonstrations, which causes oscillations and delays convergence. Switching to self-imitation learning using the imitation regulator helps reinforce some of the past good behaviors preventing the policy from diverging, which is shown by the higher mean and smaller variance of the condition 5 compared to those of condition 4.

## 6  Conclusions

We conclude that our approach, SCAPE, is capable of producing a successful state-dependent stiffness control policy, which plays a crucial role in ensuring safety and performance in dexterous manipulation. SCAPE produces competent manipulation skills by improving sample complexity with augmented position control experiences. The suboptimality of the augmented demonstrations is alleviated by a combination of the Q-filter and the imitation regulator, which results in faster and more stable convergence to a successful policy. These techniques prevent the agent from blindly imitating the suboptimal demonstrations, and help focus on the past desirable experience. Through various manipulation experiments, we validate that SCAPE outperforms the existing position control and stiffness control approaches. Therefore, SCAPE provides both safety and performance such that robust dexterous manipulation can be conveniently learned without stiffness control demonstrations. Future work may include extension of our work to seek the feasibility of passive stiffness modulation, which is a capability of human hands. Passive stiffness not only determines the safety under robot malfunction but also the dynamic behavior of the robot under sudden impacts.

**Acknowledgments**

This work has taken place in the ReNeu Robotics Lab and Personal Autonomous Robotics Lab (PeARL) at The University of Texas at Austin. Effort in the ReNeu Robotics Lab is supported, in part, by NSF (1941260, 2019704), Facebook and Dept of VA. PeARL research is supported in part by the NSF (IIS-1724157, IIS-1638107, IIS-1749204, IIS-1925082), ONR (N00014-18-2243), AFOSR (FA9550-20-1-0077), and ARO (78372-CS).

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
