# OpenReview forum: "SCAPE: Learning Stiffness Control from Augmented Position Control Experiences"
_robot-learning.org/CoRL/2021/Conference — CoRL2021 Poster_

### Official Review · Reviewer_Lq7F · 2021-07-23

**Originality:** Good
**Technical Quality:** Good
**Clarity Of Presentation:** Good
**Impact:** 4

**Recommendation:**

Weak Accept: I recommend accepting the paper, but will not argue for my recommendation if the majority of other reviewers have a different opinion.

**Summary:**

This paper introduces an RL algorithm guided by demonstrations that learns stiffness control policies for dexterous manipulation tasks involving multi-stage actions such as approach->grasp->transport->place object or make contact->move/rotate object->release contact, from position-only demonstrations. As rightfully mentioned, existing RL approaches for such dexterous manipulation tasks have focused solely on position control policies, disregarding the effect of the contact forces on the object that can damage or break it in the real-world evaluation. This can be alleviated by learning state-dependent variable impedance/stiffness control policies for object manipulation. Although there exists RL approaches for learning variable impedance control policies for contact-rich manipulation tasks, such as wiping, shoveling, peg-in-hole, etc. most of these approaches focus on a specific control policy tailored to the nominal action, not including the grasping or releasing of a tool. In this work, this is achieved by adopting the HER (Hindsight Experience Replay) + Q-filter approach [19] and augmenting it with a self-imitation learning strategy [35] which regulates the source of imitation, either from demonstrations or from past experiences. The success of the method lies in the augmentation of position-only demonstration trajectories with estimated stiffness, that although suboptimal, are capable of guiding the learning of a stiffness control policy that can minimize interaction forces while reaching task success. The algorithm was evaluated in 3 contact-rich simulation environments of varying complexity and 1 real-world experiment showcasing the transfer of a learned stiffness control policy in simulation to the real-world test bed with no extra fine tuning.

**Issues:**

Clarify issues, questions and suggestions I mention in the “Strengths and Weaknesses” section. Also, if running a new test simulation for the block environment with the pure approach from [19] is not feasible, mentioning the convergence of that paper + analysis of when the gripper exceeds breaking force on the object (as compared to SCAPE) would be sufficient.

**Reviewer Expertise:**

Very good: Comprehensive knowledge of the area

**Strengths And Weaknesses:**

The RL+demos algorithm presented in this paper has two major strengths above state-of-the-art that contribute to improving safety and scale:
(i) learning stiffness control policies using solely position trajectories while using no force measurements. This is useful as oftentimes manipulators, especially grippers or robot fingers, are not equipped with reliably force/torque sensing yet they are tasked to be used in contact-rich scenarios. Further, due to the demonstrations the robot does not learn the state-dependent stiffness function from scratch but it finds an optimal policy from the suboptimal demonstrations.
(ii) learning state-dependent stiffness control policies for multi-stage object manipulation tasks. As mentioned above, previous approaches for variable impedance control RL have failed to learn such multi-stage tasks where the manipulators make or break contacts.
The evaluation of the proposed algorithm is thorough and convincing, the paper would benefit from clarifying many important details related to the strengths listed above.

First of all, while the overall claim of not needing force measurements is compelling and useful, it gets confusing when we get to the reward function (3) and see that it needs the applied force to shape the reward. Only until reading the appendix does one discover that this force is in fact the estimated force from series elasticity of the gripper or the fingers. This estimation procedure must be highlighted and explained in the main text. Is it simply estimated by F_{est}=k_{est}\delta x? What would the displacement be? If it is computed by some displacement is it of the fingertips/joints? Furthermore, is this force estimated from the k_{passive} stiffness used for the suboptimal demonstrations? Is there a relation there? I would assume k_{est}=k_{passive} however this is not clear from the text.

Second, although the appendix properly describes the action/state spaces of each environment, the description in the main text of the learned stiffness control policies and the low-level controller is lacking. One would assume that the “controller” box in the diagram from Fig. 1a is a joint position controller that takes q_{des} and k_{des}, however is the controller a simple P-controller? PD-controller? Is this the same for all environments? Further, the structure of the policy is not discussed or the off-policy method used, is it DDPG as in [19] and the original HER approach?

Third, it would be important to discuss the type of demonstrations that this approach would work for. When using kinesthetic demonstrations, the stiffness of the robot when demonstrating the task, k_{passive}, is not the desired stiffness during task execution as this will be very low or even 0 (depending on the type of kinesthetic teaching being used either gravity compensation, impedance or admittance control). In the experiments presented in this paper, it seems that the demonstrations are given through teleoperation. However, this is not really discussed anywhere and is what I inferred from reading the related papers. It would be great if the authors clarified this and possibly discussed how this approach can be extended to other types of demonstration interfaces.

Fourth, it would be interesting to see how this approach fairs with real force/torque measurements in the demonstrations rather than the estimated ones. Would the learning converge faster? Would there be more or less oscillations when optimizing with the suboptimal demonstrations? This is important to understand the applicability of this approach to other platforms/demonstrations where force/torque measurements are available.

Fifth, the related work section is missing some relevant papers that the authors might find interesting:

1* https://arxiv.org/pdf/2004.10886.pdf

2* https://link.springer.com/article/10.1007/s10514-017-9636-y

3* https://arxiv.org/pdf/2103.15406.pdf

1* and 2* are RL approaches that learn variable impedance control policies. 3* not so relevant to this paper but might find interesting as it explores the effect of stiffness in a simulation environment on the learning parameters.

Finally, for the block environment, it would be interesting to see how much of an improvement the proposed method is as opposed to the position-control only policy approach offered in [19]. This approach is similar to Condition 4 in the ablation study, but not quite because it’s considering the safety constraints in the reward function. However, the approach in [19] is equivalent to RL+IL with non-augmented demos + Q-filter with only task-success as reward. It would seem (from comparing the results in [19]) that SCAPE is much faster than [19]-- while at the same time keeping the block safe. Does this mean that using stiffness control policies yields more efficient learning than position-control only? This would be an interesting question to answer and analyze.

+ After reading this paper, I finally understood the benefits of the HER approach.


**Summary Of Recommendation:**

This paper offers an interesting solution to a challenging problem in RL for dexterous manipulation. Hence, I believe it can be impactful but the paper needs some pruning and clarifications on methods and design choices before it is publishable.

---

> ### Author Response · Authors · 2021-08-26
> **Response to Reviewer Lq7F (Part 1)**
>
> We thank the reviewer for providing expertise and thorough feedback for our work. We are also grateful to the reviewer for the supportive comments on the overall work, and for establishing an interesting question regarding the learning curve, as well as some possible directions to further strengthen our work. We provide detailed answers and the locations of corresponding changes in the revised manuscript below. Please note that the reference numbers have changed due to the change in the main text:
>
> **Clarifications in Main Text**
>
> Thank you for these comments. As suggested, we have included the clarifications in the main text for the following:
>
> **The force estimation (Lines 182-184)**
>
> The force is estimated from the basic relationship that defines the stiffness, which is F_{est} = k_{passive}(x_{des} - x_{meas}). In the proposed settings, forces are estimated from series elasticity, therefore k would be the passive stiffness, and x would be the difference between the desired position and the current position in the respective coordinate system. For our experiments, this coordinate system is in task space (e.g., in the grasping direction for the In-hand Manipulation environment). We further note that this measurement has been filtered with a low-pass filter that has a cutoff frequency of 20 Hz. This filtered measurement is indeed used for the suboptimal demonstrations. The proposed augmentation process adds k_{passive} to the observation, and stiffness action (which is zero, since stiffness isn’t changing from the inherent stiffness) to the action, forming a proper state-action pair for stiffness control.
>
> **The lowermost level controller (Lines 119-120)**
>
> The ‘controller’ block in Fig. 1a refers to a high-level controller, which is a task-space stiffness controller in this case. The lowermost level controller, a joint-space position controller, is hidden inside this block and uses a proportional control. This applies to all the environments in our study. In the real-world environment, actuators used in the NuFingers testbed are PID-tuned to follow the desired position profile with a rise time of 0.012s. Here, the assumption is that the actuators follow the desired position faithfully. As series elasticity is used in the transmission, the behavior of the robot is dominated by the series elasticity because it is significantly more compliant than the actuators.
>
> **The learning structure (Lines 115-116)**
>
> The structures and the algorithms are DDPG + HER, similar to [11]. The only difference in the structure is the employment of the imitation regulator for self-imitation learning.
>
> **The demonstrations (Lines 128-131)**
>
> As pointed out by the reviewer, it is true that demonstrations can be provided through different means. Common ways of providing demonstrations include teleoperation, trajectory generator, kinesthetic teaching, kinematic replay, etc. Teleoperation and kinesthetic teaching are the most common approaches, but as the motion complexity increases, teleoperation tends to be easier than kinesthetic teaching (e.g., teleoperating a robot hand with a mocap glove vs. moving each joint of the robot hand manually by hand). The proposed method will work with any demonstrations as long as desired positions are provided, regardless of the coordinate frame (e.g., task space or joint space). Once the desired and current positions of the robot are known in any frame, it can be transformed to a desired frame, in which we define the stiffness. Demonstrations provided by a trajectory generator or teleoperation explicitly provide desired positions.
>
> In the paper, we used the trajectory generator such as in [11], which can also be thought of as a teleoperated robot. Other forms of demonstrations such as kinesthetic teaching, can also be used as long as desired positions can be defined. Since kinesthetic teaching only provides current positions, the desired positions can be defined either as offset from the measured positions, or from force measurements using the passive stiffness (e.g., x_{desired} = F_{meas} / k_{passive}). Once desired positions are known, the demonstrations can be augmented. Kinesthetic teaching without such mechanism (e.g., simple kinematic replay), however, has shown to be of little use in force interaction tasks as depicted in [17].

---

> > ### Author Response · Authors · 2021-08-26
> > **Response to Reviewer Lq7F (Part 2)**
> >
> > **Comparison with [11]**
> >
> > Thank you for raising an interesting question. A direct comparison between SCAPE and the results from [11] cannot be made due to the difference in not only the reward function but also the environment itself. As explained in the supplementary material, we have made several modifications to the existing Block environment. However, it is possible to remove the force penalty in the reward function, and repeat the position control policy [11] experiment in Section 5.1 to compare the learning curve of the task-related success rate.
> >
> > We have included the experiment suggested by the reviewer in the supplementary material. Please refer to the first stage of the hybrid approach, which is identical to [11]. While the results suggest [11] learns the kinematic task faster than SCAPE for both the Block and Chip environments, the process of acquiring the policy is far more dangerous as shown in Fig. 9. Interestingly, in the NuFingers environment, SCAPE learned to solve the problem quickly even with the force penalty. However, as this is not the general trend, it is difficult to make concluding remarks at this point. For our future study, it will be worth investigating how the learning curve changes depending on the weight of the force penalty.
> >
> >
> > **Force/torque Measurements (Lines 267-268)**
> >
> > Thank you for raising this very interesting point: how is learning affected when direct force/torque measurements are available (instead of force estimations from SEAs). Our approach will work seamlessly when force/torque measurements are available and we will likely see marginal improvements in situations where dynamic effects (e.g., from inertia and from high speed movements) are small. This is true in all the environments that we analyzed. In situations where dynamic effects are dominant, direct force/torque measurements will likely have significant positive impact on the learning. Because, in such situations, in the absence of direct force/torque measurement, the agent must ‘learn’ to separate the effect of external forces/torques from those arising from inertial and coriolis components.
> >
> > **Miscellaneous**
> >
> > Thank you for the list of related works that were missing. They are indeed relevant to our study, and we have included them in Section 2 accordingly.

---

> > > ### Comment · Reviewer_Lq7F · 2021-09-04
> > > **Response to Authors**
> > >
> > > I greatly appreciate the effort the authors have made in addressing all of my comments and suggestions, as well as those from the other reviewers. The revised submission is in a much better state and the inclusion of the additional experiment elucidates now the advantage of SCAPE over baseline methods. The experiments presented in the paper (for 1D and 2D stiffness control) are sufficient to evaluate the method for this submission. The validation of sim2real transfer is notable. However, for future iterations or extensions, it would be really interesting to see how this approach scales to higher task-space dimensionality and complexity. One of the major motivations of this paper is in-hand dexterous manipulation. Hence, it would be great to evaluate this approach applied to multi(>2)-finger grasping and in-hand manipulation where object-level impedance is generally used to represent/control the tasks as in the following references:
> > >
> > > - https://ieeexplore.ieee.org/document/6907861
> > >
> > > - https://arxiv.org/pdf/1909.10034.pdf
> > >
> > > It would be also interesting to see how this approach extends to more complex systems, like a hand-arm system subject to underaction as in:
> > > https://link.springer.com/article/10.1007/s10514-020-09942-9
> > >
> > > My initial recommendation of 'weak accept' remains.

---

### Official Review · Reviewer_V3q7 · 2021-07-23

**Originality:** Fair
**Technical Quality:** Fair
**Clarity Of Presentation:** Fair
**Impact:** 2

**Recommendation:**

Weak Reject: I recommend rejecting the paper, but will not argue for my recommendation if the majority of other reviewers have a different opinion.

**Summary:**

The paper proposes an insight for reinforcement learning of variable impedance control policies with only kinematic demonstrations.
The main idea is to train a policy for impedance control with behavior cloning using kinematic trajectories and the default stiffness from the model of the robot's controllers.
This provides a suboptimal impedance policy for the agent to imitate and improve upon.
Noting the shortcomings of an existing approach to RL with demonstrations (the Q-filter), the paper also proposes an "imitation regulator" that helps to trade off policy exploration with remaining faithful to demonstrations.


**Issues:**

- Damping is entirely omitted. Is everything implicitly critically damped?

Minor Issues
- Line 24-25 "in an unstructured environments"
- I had some difficulty reading the plots because I'm colorblind - please consider using a more readily readable palette.
- It might be useful to point readers to the recent review "Variable Impedance Control and Learning - A Review" from Abu-Dakka and Saveriano.

**Reviewer Expertise:**

Very good: Comprehensive knowledge of the area

**Strengths And Weaknesses:**

Strengths
- The paper presents an insight for learning from demonstration with variable impedance control - if only kinematic demonstrations are available, as opposed to the full force profile, the hand/arm's default impedance is informative as to what impedance value the agent's policy ought to be initialized with.
- The paper also proposes an improvement to the Q-filter algorithm using success rate rather than Q-value thresholds, which they observe to have an improvement on stability.
- Hardware sim2real results are successful.

Weaknesses
- The assumptions could be stated more explicitly. In particular, the method assumes that the demonstration was provided as a kinematic path and executed via the robot's "standard" position controller with stiffness k_passive. This could be the case - e.g. if the arm were naively executing commands from a  keyboard/mouse or motion planner - but there are other possibilities as well, e.g. kinesthetic teaching, or a demonstration recorded with a task-space controller rather than a joint space controller, etc. I think there are additional cases here that at least warrant mention or a clarification of the problem setting.

- I was surprised the position-controlled baseline failed the blocks task (wrt performance) given that it succeeds in the original version of the problem.

- The "series elasticity" force measurement procedure needs to be elaborated upon.
- The performance boost from the proposed imitation regulator is relatively marginal.




**Summary Of Recommendation:**

To my knowledge, the paper is one of the first to present results on learning from demonstration in the context of variable impedance control, but lacks sufficient novelty, clarity, and depth.

---

> ### Author Response · Authors · 2021-08-26
> **Response to Reviewer V3q7 (Part 1)**
>
> We thank the reviewer for providing expertise and valuable feedback for our work. We are also grateful to the reviewer for the supportive feedback on the experiments, suggestions to strengthen our work through explicitly stating the assumptions that are being made. We provide detailed answers and the locations of corresponding changes in the revised manuscript below. Please note that the reference numbers have changed due to the change in the main text:
>
> **Explicit Assumptions (Lines 128-131)**
>
> Thank you for pointing this out. As pointed out by the reviewer, it is true that demonstrations can be provided through different means. Common ways of providing demonstrations include teleoperation, trajectory generator, kinesthetic teaching, kinematic replay, etc. Teleoperation and kinesthetic teaching are the most common approaches, but as the motion complexity increases, teleoperation tends to be easier than kinesthetic teaching (e.g., teleoperating a robot hand with a mocap glove vs. moving each joint of the robot hand manually by hand). The proposed method will work with any demonstrations as long as desired positions are provided, regardless of the coordinate frame (e.g., task space or joint space). Once the desired and current positions of the robot are known in any frame, it can be transformed to a desired frame based on the model, in which we define the stiffness. Demonstrations provided by a trajectory generator or teleoperation explicitly provide desired positions.
>
> In the paper, we used the trajectory generator such as in [11], which can also be thought of as a teleoperated robot. Other forms of demonstrations such as kinesthetic teaching, can also be used as long as desired positions can be defined. Since kinesthetic teaching only provides current positions, the desired positions can be defined either as offset from the measured positions, or from force measurements using the passive stiffness (e.g., x_{desired} = F_{meas} / k_{passive}). Once desired positions are known, the demonstrations can be augmented. Kinesthetic teaching without such a mechanism (e.g., simple kinematic replay), however, has shown to be of little use in force interaction tasks as depicted in [17].
>
> **Reasons for Failure of [11]**
>
> The position control policy fails in the Block environment mainly due to the force penalty in the reward function that penalizes proportionally to the applied force. As we explain in Lines 240-241, the original version of the task is only the kinematic part. As can be seen from Fig. 3a, the task-related success rate is indeed slowly increasing even with the original approach [11], but it ultimately fails the overall task by breaking the object. Please refer to the supplementary material where we added additional experiments where the approach in [11] is used without the force penalty. As expected, the approach quickly converges to 100% task-related success rate, while failing the safety-related goal in all attempts.
>
> **Force Estimation from Series Elasticity (Lines 182-184)**
>
> The force is estimated from the basic relationship that defines the stiffness, which is F_{est} = k_{passive}(x_{des} - x_{meas}). In the proposed settings, forces are estimated from series elasticity, therefore k would be the passive stiffness, and x would be the difference between the desired position and the current position in the respective coordinate system. We further note that this measurement has been filtered with a low-pass filter that has a cutoff frequency of 20 Hz.

---

> > ### Author Response · Authors · 2021-08-26
> > **Response to Reviewer V3q7 (Part 2)**
> >
> > **Contribution of Imitation Regulator**
> >
> > The major difference between with (Condition 5) and without (Condition 4) the imitation regulator is shown in two aspects. First, the variability of the resulting policies is comparably larger without the imitation regulator. Second, the delayed convergence is noticeable in Condition 4. Condition 5 monotonically leads to a successful policy, whereas Condition 4 does not reach 100% success rate with the same amount of computation given.
> >
> > **Damping Omission**
> >
> > In our work, stiffness control alone delivered sufficient performance without damping. We expect the introduction of damping to help stabilize the control, and lead to better performance. However, adding damping as part of the policy output extends the dimension of the action space even further, and may lead to exacerbated sample complexity. Therefore, it is a common practice in most literature to adjust the damping according to the stiffness parameter so that the system is always in a critically damped state, as mentioned by the reviewer. However, such damping control requires measurements of velocity in the joints, and/or additional hardware to control damping. So we decided to omit damping since stiffness control alone provided sufficient performance.
> >
> > **Miscellaneous**
> >
> > Thank you for these suggestions. We have modified the line styles to improve readability, corrected the errors, and included the review paper as part of the reference as it deeply relates to our study.

---

### Official Review · Reviewer_jxpB · 2021-07-24

**Originality:** Good
**Technical Quality:** Very Good
**Clarity Of Presentation:** Very Good
**Impact:** 3

**Recommendation:**

Weak Accept: I recommend accepting the paper, but will not argue for my recommendation if the majority of other reviewers have a different opinion.

**Summary:**

As previous works have discovered, variable impedance control is an effective action space for RL in robotic manipulation. However, for many tasks, learning a good policy without an initialization trained on expert demonstrations is difficult; furthermore, collecting demonstrations in variable impedance space is considerably more complicated than simpler position control demonstrations.

To solve this issue, the authors propose SCAPE, with the following key contributions:
1) A method for converting expert position control demonstrations to suboptimal variable impedance demonstrations, by inferring stiffness from the position controller’s feedback gains.
2) A novel “Imitation Regulator,” which allows for stable switching from demonstration to the robot’s prior experience in the replay buffer.
3) A series of comparative and ablative experiments demonstrating the method’s superiority over a) position control (fixed impedance) with demonstrations and b) variable impedance control without demonstrations, in which the proposed method outperformed the baselines.



**Issues:**

Addressed major comment:
Authors are recommended to bolster the motivation of the project in the paper, particularly by elaborating on why converting position control demonstrations is better than collecting demonstrations in the proposed action space.

Addressed Nitpicks:
1) The authors occasionally make some conclusive statements without sufficient evidence. For instance, that another method’s poor performance “is due to the high dimensionality induced by the variable impedance that 99 hinders policy search” (lines 98-99). It is also not clear to the reader how the spikes in force on the real-world nu-fingers can be attributed to “poor force tracking in MuJoCo” (line 263).
2) The difference in scale between the forces generated by the baseline & SCAPE in the NuFingers demo make the plot (Fig. 4c) difficult to read.
3) It is not clear to the reader why the authors refer to their method as “stiffness control,” while all of the cited related methods ([29, 30, 31]) take the convention of calling the action space “impedance control.” I suggest either elucidating the difference between the two, or using the more standard name if there is none.


**Reviewer Expertise:**

Good: General knowledge of the area

**Strengths And Weaknesses:**

Strengths
1) The proposed method displayed significant performance gains over properly selected ablations and comparisons. The increase in data efficiency and safety provides a compelling argument for using variable impedance control with demonstrations when they are available. It was particularly great to see real-world experiments to this end.
2) The explanation of the method was clear and concise.

Addressed Weaknesses
1) A central motivational point of the paper is that expert demonstrations where impedance is being controlled explicitly are “expensive and difficult to acquire” (line 38). It’s not very clear to the reader where this difficulty comes from, as it is only briefly mentioned in the intro.


**Summary Of Recommendation:**

The method clearly gives significant gain in data efficiency, performance, and safety to the presented tasks. However, I don’t see much future work being spurred by SCAPE.

---

> ### Author Response · Authors · 2021-08-26
> **Response to Reviewer jxpB**
>
> We thank the reviewer for providing expertise and valuable feedback for our work. We are also grateful to the reviewer for the supportive comments on the real-world experiments, as well as the helpful suggestion to strengthen the motivation of the study. We provide detailed answers and the locations of corresponding changes in the revised manuscript below. Please note that the reference numbers have changed due to the change in the main text:
>
> **Study Motivation (Lines 35-41)**
>
> Thank you for the suggestion to bolster our motivation. Commonly, expert behavior is measured directly using various sensors. However, stiffness cannot be measured directly since it is a relationship, rather than a measurable quantity. For example, even with measured position and force profiles, the number of possible combinations of desired position and stiffness profiles is infinite.
>
> A common way of collecting stiffness data is shown in studies such as (The Control of Hand Equilibrium Trajectories in Multi-Joint Arm Movements, T. Flash. The central nervous system stabilizes unstable dynamics by learning optimal impedance, E. Burdet, et al., A method for measuring endpoint stiffness during multi-joint arm movements, M. Kawato). First, admittance control is implemented to capture equilibrium position trajectories. Along the trajectories, the demonstrator is purposefully perturbed with subtle and quick impacts to measure compensatory forces/torques, which are then used to implicitly calculate the stiffness at that position (F = kx). A set of stiffness estimates at certain positions are then used to further estimate the stiffness profiles, thereby compounding potential errors. This process is more ambiguous for object manipulation due to the required precision and accuracy especially for in-hand manipulation.
>
> To summarize, collecting expert demonstrations for stiffness control not only requires specialized hardware, but also consists of meticulous procedures where errors in estimation could be significantly detrimental. This is one of the major challenges to learning stiffness control from demonstrations. As can be seen from our ablation studies, however, multi-stage manipulation tasks are difficult to solve without these demonstrations. Therefore, we propose a way to learn Stiffness Control from Augmented Position control Experiences (SCAPE).
>
>
> **Term Convention**
>
> Thank you for raising this question. The impedance consists of mass, spring, and damper, and dictates the response motion of the object to the mismatch between the desired trajectories and observed states by indirectly modulating applied forces. However, it is common to exclude the mass part due to the difficulty of measuring the acceleration directly. Therefore, researchers often employ impedance control with ([24,25,27], compliant control) or without (ours, stiffness control) critical damping. However, critical damping requires additional estimation or measurement of velocity, but we have found that our approach is successful even without critical damping. Therefore, we have decided to follow the convention and use the specific term stiffness control since damping is not included in our study.
>
>
> **Miscellaneous**
>
> Thank you for the suggestions. We have revised the manuscript and edited some conclusive statements as well as the scale for the force plot in Figure 4c. We would also like to clarify that Figure 4c is generated during the training in the MuJoCo environment, and that the policy is directly transferred to the real-world environment without additional training. Therefore, the force spikes shown in Figure 4c are generated on MuJoCo, which could arise from the unstable contact between two approximated convex surfaces (i.e., the object and the fingertips). In the real-world environment, however, such spikes are not shown, as can be seen from the video in the supplementary material.

---

### Official Review · Reviewer_JwaB · 2021-07-30

**Originality:** Good
**Technical Quality:** Very Good
**Clarity Of Presentation:** Very Good
**Impact:** 3

**Recommendation:**

Weak Accept: I recommend accepting the paper, but will not argue for my recommendation if the majority of other reviewers have a different opinion.

**Summary:**

This paper presents SCAPE, an approach for learning stiffness control from standard position control demonstrations by augmenting them with the inherent passive stiffness of the robot. This demonstration augmented with the inherent stiffness, which can be obtained from the model or the physical specs, can serve as a sub-optimal demonstration of a stiffness control trajectory. SCAPE learns a position control policy by imitation learning from these demonstrations; in order to avoid learning from highly sub-optimal actions an additional Q-filter is used. This Q-filter switches learning between the demonstration and the robot’s actions at a given state depending on their relative Q values. To further reduce oscillatory behaviour when learning with such a switching loss and to encourage exploration, an additional imitation regulator is used. This regulator switches to learning completely from self-experience once the policy starts achieving significant success thereby allowing the policy to bootstrap from its own experience towards the end of learning. The approach is tested on three robot manipulation tasks in simulation and the real world and shows good performance compared to baseline methods, both in terms of final performance and obeying safety constraints. An ablation also evaluates the different parts of the proposed approach.

**Issues:**

Further experiments are needed as discussed above. More details on the experimental setup and ablations can also add strength to the paper.

**Reviewer Expertise:**

Good: General knowledge of the area

**Strengths And Weaknesses:**

This paper proposes an approach to leverage position control demonstrations to learn stiffness control policies by demonstration augmentation. The approach is evaluated on three manipulation tasks and shows good results compared to a baseline method. The ablation study also provides useful insights on the usage of the different newly proposed components in this context. A few comments:
1. The proposed approach needs position control demonstrations to bootstrap policy learning. How many demonstrations are needed for stable learning? How many demos are used for the results presented in the paper? It would be useful if these numbers are provided and an additional ablation is run w.r.t the number of demonstrations used for training.
2. While it is mentioned that baseline methods cannot use imitation learning since they do not have access to the augmented demonstrations, it would be interesting to see if the imitation learning loss is used just to optimise the action dimensions while the online RL is used to optimise the stiffness actions. This would be a hybrid approach where an initial policy would be trained with imitation learning (this would only be in the standard action space) and an additional policy could be trained on top of this policy using RL which sets the stiffness actions. This would be a better and fairer comparison compared to the baseline methods as it does have access to the demonstrations but not the augmented stiffness component.
3. It looks like the presented baseline approach does pure position control and does not actually do any stiffness control. Is it possible to compare against a baseline that learns stiffness control from scratch on the proposed tasks? This would also give a clearer picture of the “near-optimal” stiffness control policy as by learning from the scratch the policy would not be biased by the sub-optimal demonstrations. It would also shed light on how well SCAPE does in terms of performance compared to a near-optimal policy.
4. Can the approach be extend to multi-joint stiffness control? What would be the main limitations in this regard? A discussion on this would be useful.
5. The approach uses the inherent stiffness of the manipulator to augment the position control demonstrations. What would happen if there is noise in this stiffness parameter, i.e., there is mismatch between the used stiffness and the actual stiffness of the system? Were any experiments done with noise added to this stiffness parameters? Is SCAPE able to recover a good policy even in this setting?

**Summary Of Recommendation:**

Overall, the proposed approach is well motivated and shows good results across the tasks it is evaluated on. That being said the baselines seem weak and more analysis and comparison is needed to clearly establish the strengths of the method.

After rebuttal: I've increased my score to a "Weak Accept" as the rebuttal addressed several of my concerns.

---

> ### Author Response · Authors · 2021-08-26
> **Response to Reviewer JwaB (Part 1)**
>
> We thank the reviewer for providing expertise and insightful comments on our work. We are also grateful to receive the supportive comments on the main and ablative experiments and for suggesting a stronger baseline. Below are detailed responses, including pointers to changes in the manuscript. Please note that the reference numbers have changed due to the change in the main text:
>
> **Additional Baseline (Supplementary Material)**
>
> Thank you for the suggestion. We would first like to clarify that the baseline approach for the position control policy in Section 5.1 indeed uses identical demonstrations as SCAPE, except the augmented stiffness part. The only baseline method that does not have access to the demonstrations is Condition 1 in Section 5.2, which learns stiffness control from scratch as in [24,25,27].
>
> The hybrid approach suggested by the reviewer is similar to [21, 22], except that here we assume that we do not have access to the expert trajectory generator. We agree with the reviewer that this experiment would add value to our work. Therefore, we have added the experiments and the results indicate that SCAPE outperforms this baseline too.
>
> Based on imitation learning alone, the agent could not learn the task due to the small number of demonstrations provided (25). Thus, we use IL+RL (position) → RL (stiffness). For the (IL+RL) stage, we have removed the force penalty in the reward function as in [11], so that the agent can learn to grasp the object before the next stage starts. The total number of timesteps matches with SCAPE for fair comparison. For the In-hand Manipulation environment, however, the agent was not able to find such policy within the available timesteps. Therefore, for this environment, we assign twice as many timesteps as we assigned to SCAPE.
>
> The results suggest that despite the kinematically successful controller from the (IL+RL) stage, the agent still fails to learn safe stiffness control. For the In-hand Manipulation environment in particular, it is interesting to see that SCAPE still outperforms at such disadvantage. We postulate that without the force penalty, the agent focuses on the kinematic task only, moving and grasping as quickly as it can without considering the interaction force. Therefore, even if we train the agent further in the stiffness control domain, the agent still possesses this characteristic, leading to failure. So, we conclude that SCAPE outperforms the hybrid approach as well. Also, due to the ambiguity in determining the number of timesteps for each stage of learning, and the severely deteriorated safety during policy improvement, we include this analysis only in the supplementary material.
>
>
> **Baseline Learning Stiffness from Scratch**
>
> Thank you for the suggestion. We want to clarify that this is shown in the ablation study in Section 5.2. Condition 1 refers to the approach that learns stiffness control from scratch as suggested by the reviewer. This approach is also taken by the past studies [24,25,27], and the low success rates confirm that these methods do not succeed in multi-stage tasks demonstrated in our paper.
>
> **Number of Demonstrations (Lines 128-131)**
>
> We used 25 position control demonstrations in all the experiments. For SCAPE, we augmented the identical demonstrations that are used for the position control baseline. From our experience, varying the number of demonstrations did not have a significant impact on the learning curve unless it is too small as mentioned above for imitation learning. Therefore, we did not include additional analysis of varying the number of demonstrations.

---

> > ### Author Response · Authors · 2021-08-26
> > **Response to Reviewer JwaB (Part 2)**
> >
> > **Applicability to Multi-DoF (Lines 139-141)**
> >
> > SCAPE is not limited to single-DoF stiffness control. The core idea of our paper is to infer the stiffness from the model and make the position control demonstrations appear to be stiffness control demonstrations. Therefore, as long as the model is known, the number of available adjustable parameters would be n for n-dimensional stiffness. In fact, for our In-hand Manipulation environment, the model provides 2D stiffness in the task space, since it was 2D planar manipulation. However, we chose to use only the direction that is relevant to our problem, reducing it down to 1D stiffness (only in the grasping direction). If the problem also concerns the rotational torque in addition to the grasping force, then we simply include the full 2D stiffness in the observation and action spaces.
> >
> > **Noise in Stiffness Parameter**
> >
> > Thank you for raising an interesting question. In our study, agents experience noisy apparent stiffness, caused by different types of uncertainties including the observation noise, random perturbations, and control failures. Here it is crucial to distinguish the terms ‘apparent stiffness’ and ‘commanded stiffness’. In a practical setting, the commanded stiffness is not affected by sensory feedback, but rather determined as a feedforward parameter. Therefore, in the observation from the agent side, there is no noise included in the stiffness parameter itself. As pointed out by the reviewer, however, it is possible that the apparent stiffness of the robot could be noisy due to the uncertainties mentioned above. This apparent stiffness is different from the commanded stiffness since the former is the actual stiffness seen from the environment, and the latter is the stiffness commanded by the policy. Therefore, we expect the apparent stiffness to be noisy and oscillatory. Despite the noisy apparent stiffness, the results suggest that the SCAPE successfully handles such uncertainties.

---

> > > ### Comment · Reviewer_JwaB · 2021-09-04
> > > **Thanks for the rebuttal**
> > >
> > > I would like to thank the authors for their clarifications in the rebuttal and for running additional experiments to address some of my concerns. It is good to see that the presented approach outperforms a stronger baseline of IL+RL for position control and RL for stiffness control even when the baseline is given a higher learning budget.
> > >
> > > With regards to noise in the stiffness parameter I don't think the authors have quite understood my question. While it does make sense that the "apparent stiffness" on the robot is different from the one in the datasheet it is not clear what the scale of this difference is. On the other hand, one could easily perturb the stiffness parameter from the datasheet by say 10%, 20%, 30% or so and rerun the experiment to see where SCAPE breaks -- this would quantify the robustness of the algorithm somewhat (atleast on the tasks explored in the paper) and can provide a way for addressing cases where the stiffness in the datasheet poorly matches the underlying system. This could be a useful addition to the paper.
> > >
> > > Overall, I am happy with the responses provided as they address many of my concerns and will increase my score to a "Weak Accept".

---

### Meta-Review · Area_Chair_Q43v · 2021-08-11

**Recommendation:** Accept (Poster)
**Confidence:** 4

**Metareview:**

Strengths:

** The paper focuses on an important and interesting problem, learning for stiffness/force control. The experimental focus here highlights the distinction between what is possible w/ position control only versus stiffness control. This can be hard to do and I think the paper does it well.

Weaknesses:

** Unclear and weak baselines: it is not completely clear what is the baseline reported in section 5.1. Sect 5 says “we compare our results with position control (existing approach) so that the difference only lies in the policy parameterization”. I’m assuming this means that the baseline algorithm cannot change the stiffness from the passive value. Would it be possible to compare to a baseline that is identical to SCAPE except for the addition of the methods proposed in sect 3 (i.e. the same action space) or is that what you’re doing already? Also, a reviewer asks for a comparison to a method that uses IL to learn a position policy and RL to learn a stiffness policy. Generally, it seems that stronger baselines would be appropriate here.

** Important details are missing in the main text of the paper. For example, what exactly is the RL algorithm that is augmented with the ideas in sect 3 and what is the baseline algorithm? I believe it is DDPG + HER, but this should be explicit. Loss functions would help here. How were demonstrations obtained and how many were used?

** Some of the reviewers were unclear about the high level motivations of the paper. It would also help to frame the problem more formally, i.e. clarify the assumptions about the absence of stiffness demonstrations and why stiffness demonstrations are hare to obtain. I think the contributions could be stated more explicitly also: the basic methods proposed in sect 3 come from the literature (maybe IR is the key contribution?). It seems the main contribution here is in the application. So, why do we refer to this as a new method, i.e. “SCAPE”?

Post-rebuttal:

The authors have made significant improvements to the motivation and explanations of key points identified by the reviewers. The authors have also added additional experimental comparisons that help understand the method better.

---

> ### Author Response · Authors · 2021-08-26
> **Response to Area Chair Q43v**
>
> We thank the Area Chair for providing expertise and overviewing the reviews. Here is a quick summary of our revisions. You may find more detailed responses to reviewers’ comments below:
>
> **Additional Experiments**
>
> Following the suggestions by Reviewers JwaB and Lq7F suggested we carried out additional experiments for baseline comparison. The results indicate that SCAPE outperforms the suggested baseline.
>
> **Contributions of the Study**
>
> The core contribution of SCAPE is in the exploitation of the robot model so that state-dependent stiffness control for multi-stage tasks can be learned without actual expert demonstrations. We have attempted to highlight the significant contributions by stating the challenges involved in collecting stiffness control demonstrations in the response to the reviewers.
>
> **Clarifications**
>
> We show comparisons of SCAPE with a position control policy (Section 5.1) and with stiffness control policies (Section 5.2). We have also explicitly stated the learning structure (DDPG + HER) and the assumptions on the required demonstrations as suggested by the reviewers.

---

### Decision · Program_Chairs · 2021-09-13

**Decision:**

Accept (Poster)

**Comment:**

Strengths:

** The paper focuses on an important and interesting problem, learning for stiffness/force control. The experimental focus here highlights the distinction between what is possible w/ position control only versus stiffness control. This can be hard to do and I think the paper does it well.

Weaknesses:

** Unclear and weak baselines: it is not completely clear what is the baseline reported in section 5.1. Sect 5 says “we compare our results with position control (existing approach) so that the difference only lies in the policy parameterization”. I’m assuming this means that the baseline algorithm cannot change the stiffness from the passive value. Would it be possible to compare to a baseline that is identical to SCAPE except for the addition of the methods proposed in sect 3 (i.e. the same action space) or is that what you’re doing already? Also, a reviewer asks for a comparison to a method that uses IL to learn a position policy and RL to learn a stiffness policy. Generally, it seems that stronger baselines would be appropriate here.

** Important details are missing in the main text of the paper. For example, what exactly is the RL algorithm that is augmented with the ideas in sect 3 and what is the baseline algorithm? I believe it is DDPG + HER, but this should be explicit. Loss functions would help here. How were demonstrations obtained and how many were used?

** Some of the reviewers were unclear about the high level motivations of the paper. It would also help to frame the problem more formally, i.e. clarify the assumptions about the absence of stiffness demonstrations and why stiffness demonstrations are hare to obtain. I think the contributions could be stated more explicitly also: the basic methods proposed in sect 3 come from the literature (maybe IR is the key contribution?). It seems the main contribution here is in the application. So, why do we refer to this as a new method, i.e. “SCAPE”?

Post-rebuttal:

The authors have made significant improvements to the motivation and explanations of key points identified by the reviewers. The authors have also added additional experimental comparisons that help understand the method better.